

# Subsea cable management: Failure trending for offshore wind

Charlotte Strang-Moran[1]

[1]Offshore Renewable Energy Catapult, Glasgow, G1 1RD, Scotland

*Correspondence:* charlotte.strang-moran@ore.catapult.org.uk

**Abstract.** Subsea power cable failure is an issue which is detrimental to both export cables for Offshore Transmission Owners (OFTO) and inter-array cables for wind farm operators. As the offshore wind sector advances in technology, size and capability, future sites will be farther offshore to harness the most powerful of wind conditions. As technology adapts and offshore wind develops, subsea cables are also required to acclimatise and become more reliable. This paper will review current

subsea cable failures in the UK sector. In addition, it will provide an overview of the methodology used to initiate the failure trending, and further discuss the importance of accurate data and the constraints on the initial findings.

For the purposes of this report the following definition of terms have been identified in table 1.

**Table 1: Definition of terms**

| Term | Definition |
| --- | --- |
| Cable system | Subsea power cable and cable accessories connecting to a point to which power is delivered |
| Conductor | Cable core designed for transmission of electric current, made of copper or aluminium |
| Joints | Accessories that provide connection between 2 cable ends. |
| Termination | Connections for cable, equipment or panels |
| T-Junctions | Accessory used to cover electrical 3 way connections. |
| Scour | Erosion of the seabed caused by forces from currents and waves, resulting in movement of sediments |
| Fault | abnormity of cable systems integrity and function, interrupting the operation of equipment, causing the system to fail |
| Failure | complete outages of electrical supply from a cable system |
| Root cause | initiating cause of either a condition or a causal chain that leads to a failure |
| Failure mode | identification of how the failure was primarily instigated |

## 1. Introduction

The UK offshore wind sector has currently 8.4GW of fully operational wind farms with a further 14.8GW in the pipeline. Reliable subsea cables are essential to optimise the operation of offshore wind. Improving the reliability of subsea cables, will

in turn enhance the availability of the offshore wind farm, and through this, a potential reduction of the overall Levelised Cost of Energy (LCoE).

ORE Catapult have been involved in stakeholder engagement since 2015 regarding cable failure in the UK. From these discussions, there is consensus from stakeholders, including operators and developers within the offshore wind sector that subsea cable project activities are currently siloed. There is awareness across the industry that a more structured and co-ordinated process is required to pull promising technology and ideas through development and demonstration. There has been previously no incentive to investigate certain faults and to monitor cable conditions; not all systems use online monitoring

technology. Offline testing can be expensive due to the steps involved in the procedure, as well as the required generation downtime when testing offline. Instead faults have been left to become failures, where the repair costs are covered by the insurer. The industry is adapting and shifting to a more pro-active approach through understanding where failures appear and detecting faults much earlier to allow for preventive maintenance.  The understanding of how failures develop will give stakeholders a better understanding of how monitoring technology can be effective and what cable components are particularly

sensitive.

The UK has 68 export cables totalling 1,682km, and 2,152km of inter-array cables transporting 7,990MW of generation, with currently over 50 failures recorded for fully commissioned sites. Understanding the cable market can complement failure trends and can determine the effect that a failure can cause on the wind farms' generation and performance. Using this data, in coordination with event records, the industry can understand what failures have occurred, those to be avoided, and where

innovative concepts should be encouraged.

The progression and development in subsea cable technologies, as well as understanding safer offshore fault-finding procedures can only be attained through fully recognising the causes of cable failure and the appropriate lessons learned. There is limited existing data on cable failures and there has been no effective concept to benchmark them. This has led to the development of a trend analysis database, which catalogues failures from deployed subsea power cables in UK offshore wind

farms. Analysis of the database include lifecycle factors of subsea cables, cable technology and logged failures.  Understanding cable failure trends will drive innovative technology and concepts that can lead the sector to improve efficiency and in turn reduce the financial costs associated with failures.

## 1.1  Current Trends

Subsea cables in the UK offshore wind market are both Medium Voltage Alternating Current (MVAC) and High Voltage

Alternating Current (HVAC), for inter-array and export systems respectively. MVAC cable refers to 33-66 kV and HVAC refers to any rating above 132kV, to the maximum export rating in the UK, 220 kV. Cable systems consist of the cable lengths, joints, terminations and switch gears. Currently in the UK all cable systems in operation utilise AC systems.

OFTO systems will potentially be HVDC in the coming years. ABB have been awarded contracts to supply HVDC technology for sites; Creyke Beck A, Creyke Beck B, and the Teesside A, located in the Dogger Bank cluster (Foxwell n.d.). This change

in cable technology is due to the requirement for offshore wind farms to move farther from shore to harness untapped areas of wind resource. Due to technology progression in increased rotor size and enhanced capacities (IEA n.d.), the power capabilities of cables are also expected to increase.

There is an industry drive to move towards offshore meshed grids through projects such as the Progress on Meshed HVDC Offshore Transmission Networks (PROMOTioN) project (Radowitz 2019). As the distance from shore increases, AC

technology will no longer be a viable option for exporting generation as the significant costs associated with reactive power compensation would be unattainable. Subsea HVDC technology is the leading viable alternative that is already well-tested, through Germany's wind farm transmission network and cross-country interconnectors for international exports of power.

### 1.2 Importance of Data Sharing

The purpose of the presented findings is to support the development and implementation of technologies and processes that

can reduce cable failures within the offshore wind industry, thus improving reliability of offshore wind. Utilising this method of knowledge sharing for lessons-learned from subsea cable failure could speed up the development of technological advancements, while breaking down barriers that exist for data-sharing of former and existing cable failures.

Although the cable supply chain includes professionals from manufacturing, service providers and academia, there is limited communication around improving cable failure through data sharing. The few sources that are available come across the same

challenge; lack of data straight from the source. Consensus from ORE Catapult engagement with stakeholders is that cable failure information is limited. Learned experiences can be used to forecast or evade future failures in addition to bridging innovation gaps in manufacturing and installation, whilst still maintain a level of confidentiality.

### 1.3 Accuracy

There is a perception within the offshore wind sector that subsea cable activities are ineffectively shared, and a more proactive

and collaborative direction will be required to pull innovative technology to commercial status. Incorporating big data sets to understand cable failure trends can provide a more complete and accurate perspective on assumed opinions and challenges. Working with larger data sets minimises risks of incomplete data and reduces misinformed decision-making. To capture this quantity of data the creation of a unique platform is required to provide a secure and reliable centre for trended knowledge. Until recently, proposed concepts of developing such a subsea cable platform, to inform stakeholders of gaps in innovation,

have been met with hesitation from a number of asset owners and operators. Stakeholder engagement clarified that there were reservations over whether a live, regularly-updated platform would provide ongoing value and longevity. As the industry is



changing and data trend analysis is recognised as an effective method to bridge certain technical challenges in the industry, more opportunities are materialising for accurate data-sharing. The subsea cable market will inevitably benefit in this regard.

### 1.4 Anonymity

Subsea cables across various sites can possess different characteristics, from design, manufacturer and even techniques of installation. Due to the individuality of each site specification, care must be taken when presenting fault-related data to ensure that competitors are unable to retrospectively presume the site or asset that the fault relates to. This is a limitation which constrains what data can be disseminated.

Through communication with various stakeholders, there is agreement that operators, OFTOs and the corresponding wind
farms must remain anonymous throughout the trending. Anonymity has in the past provided the basis for data sharing and has been upheld as a reliable and accurate industry process such as the System Performance, Availability and Reliability Trend Analysis (SPARTA) platform that supports advances in the reliability and performance of offshore wind farms through anonymised data (Offshore Renewable Energy Catapult n.d.). Through correctly anonymising the data, there is virtually no risk of stakeholder data security breaches, industry bias and assumptions of the trended results.

## 2.   Methodology

The trend analysis database created by ORE Catapult was created with the purpose of identifying the main drivers of subsea cable failure. The first design objectives were to understand the lifecycle of subsea cables with the purpose of the key trends being utilised by engineers and academia. The database collected selected data from wind farms such as cable capacity, conductor types, voltage rating and insulation type to provide an outline of the current specifications of the subsea cables. The
next objectives were to capture the failures of subsea cables from both private and public sources, while maintaining anonymity of the wind farm sites involved.

ORE Catapult has utilised its established relationships with key stakeholders, including owners and operators, to understand the consensus across the industry around cable failure. In addition to stakeholder engagement and industry knowledge, the collated data includes events that are available through public reports such as Ofgem and recognised sources such as 4C
Offshore. Across these publicly available sources, there were limited detailed descriptions, as well as a limited quantity of recorded events, due to the confidential nature. Without the additional input of data from industry contacts, the available sources were ineffective to trend and impossible to remain anonymous.



## 3. Results

The following results are collated from ORE Catapult's trend analysis database.

### 3.1 Failure Modes

Failure modes for subsea cables are determined by how the failure was initially caused. Subsea cables have varying risks associated with the operating environment and ways in which they can fail, and therefore experience multiple failure modes. As illustrated in

Figure 1, there are 4 main failure modes that have been collated from the recorded events in the platform. A substantial portion of cable failures documented were instigated by incorrect installation of the asset, or irregularities that developed as the asset was under the process of manufacturing. In comparison, 15% of failures were caused by ineffective design of the cable, and another 8% caused by external damages. By 2019, the collated failures for subsea cables showed that failures on components made up 30% of both manufacturing and installation failure modes.


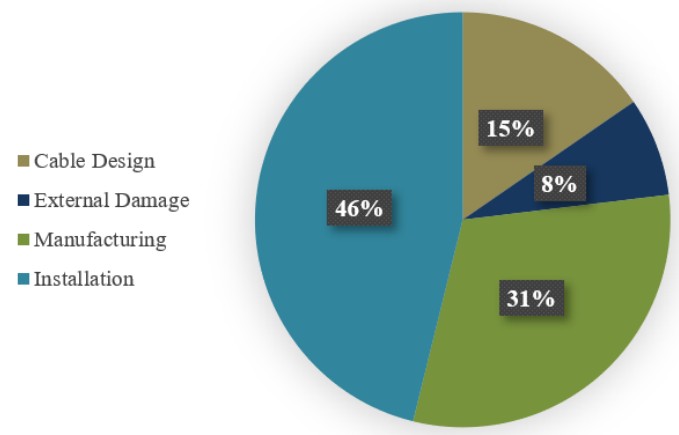

**Figure 1: Recorded failure modes for export and inter-array**

### 3.1.1 Cable Design

Inadequate design of the cable specifications was categorised as the reason for failure for 11% of findings. The cable system design also includes additional components such as connectors, joints and terminations. Poor design combined with excess sheath currents can give localised hot spots. These hotspots overheat when under pressure during operation. Due to the nature of how components connect to the cable, these can be high risk areas and will lead to failure of the asset. Components being





unsuitably selected for harsh offshore conditions, have resulted in flash-overs occur that can be dangerous to personnel who,
consequently, investigate and replace them when these faults occur.

### 3.1.2 Manufacturing

A fifth of documented inter-array failures in the database occurred due to existing manufacturing defects, that caused latent faults. Almost 40% of the recorded export cable failures across the UK were reported as manufacturing faults. An interference during manufacturing can cause a void to occur in the insulation or create holes within the insulating plug that can lead to
possible partial discharge and the breakdown of the cable. Another common fault from the manufacturing process that has been recorded, is storing or coiling the internal materials incorrectly prior to the production of the insulation and armour protection, causing damage to the fibre optics that can in turn cause breakdown of the cable.

### 3.1.3 Installation

Across the lifetime of subsea cable systems, installation errors can cause concealed faults that can go undetected at the time of
incident and will create issues within the system during operation, similar to the effect of manufacturing defects. 46% of failure modes recorded showed that cable systems had failed from faults that had originally occurred during the installation process. Failures categorised as an installation failure mode includes equipment being handled outside of their specifications, or insufficient burial of cable. Similar to risks surrounding the manufacturing process, cable system component integration is an area of high risk, making up one third of the total installation failures across both inter-array and export cable systems. Issues
occur from incorrect component integration such as mechanical stress on components, for example, over or under tightening T-Connectors.

### 3.1.4 External Damage

External damage only makes up only 8% of the total recorded failure modes. These were primarily due to instability from surrounding conditions. Scour was responsible for half of these recorded failures, occurring on the cable when there is too
much interaction between the cable and waves or current.

### 3.2 Root Causes

Examples of root causes have already been mentioned as causes of failure modes. Electrical faults induced by fibre optics have been recorded as one of the major root causes, affecting primarily export cables (15% of export failures). These faults can be unnoticed until a failure occurs. This root cause can be instigated through:
- incorrect loading of the cable
- incorrect coiling during or after manufacturing
- improper handling of the cable during installation





Damage to the fibre optic cable leads to a breakdown in the insulation that can in turn lead failure from the power core.

Ageing can cause the cable surface to become brittle, crack and eventually result in failure of both the insulating and sheathing materials. This will expose the conductor which can then potentially cause the cable to short circuit. Voids in insulation can also cause the insulating material to breakdown. Voids can occur from manufacturing if the insulating material is disturbed by even the smallest object, like a single hair or a speck of dirt. The root causes of installation failures are, for example, stress on components cable sealant and compounds from incorrectly connecting components to the system.

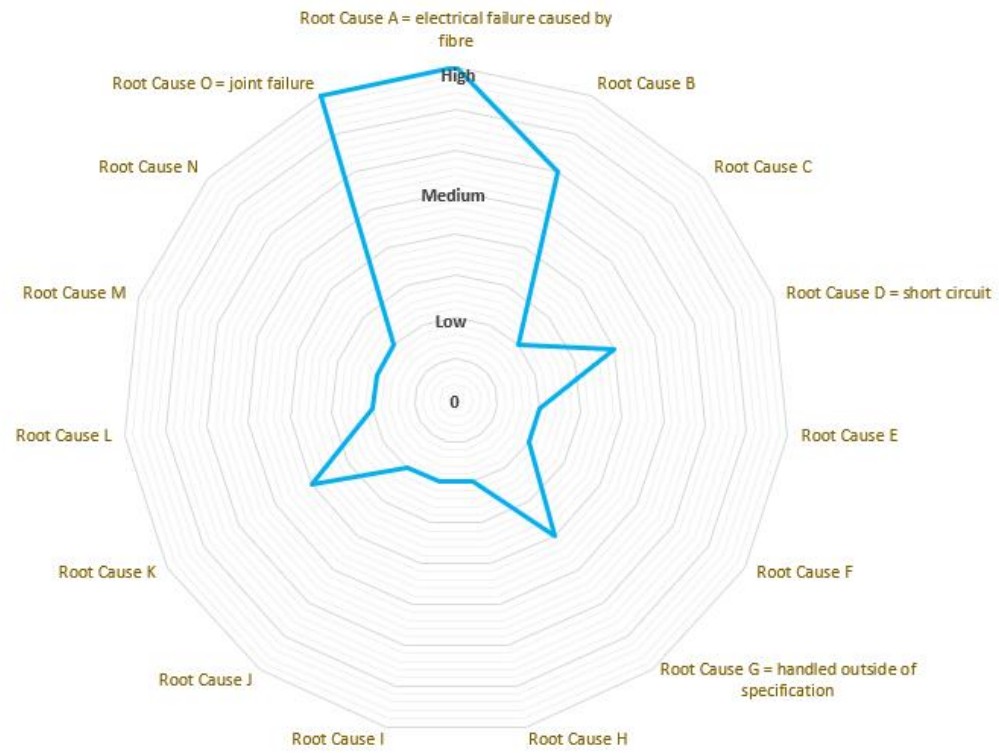


**Figure 2: Root causes from collected data**

Figure 2 only reveals just a few of the root causes that have been trended in the collated data. As there is only limited data available the anonymity of certain projects could be at risk if all root causes are revealed. As discussed previously, the revealed

root causes show that electrical failure induced by fibre optics cable is a major cause of failures. As the dataset continues to increase, the data that can be anonymised also increases allowing better analysis to be carried out.



### 3.3 Generation Downtime

The harsh conditions for offshore assets can not only have an impact on failure rates, but also affect the repair times of failures,
thus causing variation in the resulting downtime. Estimations of the total generation downtime were calculated from the collated data, the Levelised Cost of Energy (LCoE) and the annual system availability per year. When calculating the generation downtime per inter-array cable, the topologies of each individual wind farm were used to determine which cable failed and to determine if redundancy was available. The cumulative results are shown in Figure 3. Where downtime data was limited, the average duration was utilised. The average downtime for an inter-array cable failure and repair was 38 days. The
average downtime for export cable failure and repair was 62 days.

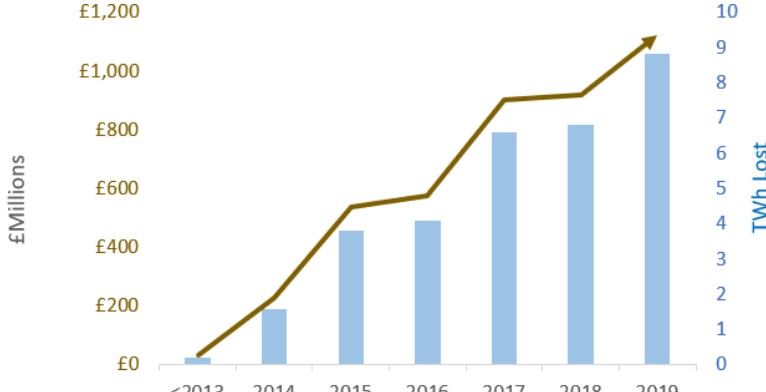

**Figure 3: Generation lost from cable failure**

The data collated and quantified in Figure 3, illustrates how failure data can be utilised to show the potential ramifications of cable failure and the almost linear increase in total TWh lost. With the approximated ferocity in cable failure, around £1,200 million has been lost just through generation losses alone. Many reasons can influence the length of downtime. This can be because the repair procedure is complex; mobilising vessels, procuring spare cable joints and accessories; and fault finding can also be a lengthy process. There are technologies out there to help reduce the downtime, but these can be expensive and
time-consuming to install.

### 4. Discussion

Other areas within the renewable sector, such as wind farm performance reliability, have been tracked successfully by the previously mentioned SPARTA and, for land based assets, the Wind Energy Benchmarking Services (WEBS) (Offshore Renewable Energy Catapult n.d.) which also similarly benchmarks performance credentials. The development of cable failure
data collection will continue to grow and mature, in data sets and available trends. By further developing discussions with key





stakeholders, including owners and operators, these first steps of trending will push forward innovation and collate industry knowledge to a singular platform.

Inter-array cables being underrepresented in logged failures is possible. Since export cables are rated at transmission voltages and supply the entirety or a substantial percentage of the generated power through one cable, there is more incentive to report

cable outages as the government regulator, Ofgem must be informed if there is any major shortages in generation. Inter-array cables can fail while not interfering with the entire supply, depending on the location and redundancy of a site. It is likely that as the dataset continues to grow there will be a shift in the current ratio of export and inter-array cable failures and in the percentages representing the listed failure modes.

### 4.1 Cable Design

In most design cases the cable insulation is expected to be ethylene propylene rubber (EPR) or cross-linked polyethylene (XLPE), both insulation materials are suitable, electrically and mechanically. HVAC subsea cable has three cores of copper or aluminium and fibre optics are strategically placed within the cable. Armour wiring is used to protect the internals of the cable against tears and snags from the installation process. The cable design or components being unsuitably selected for the harsh conditions of an offshore wind farm can be the reason of components failing during asset operation.

### 4.2 Manufacturing

The manufacturing process is intensive and requires the upmost quality to ensure the cable is to a high standard. Dormant faults can go undetected at the time of incident and create issues within the system during operation. Defects can occur due to even the slightest interference within the manufacturing process, this could be from the insulation not setting at the required temperature or a substance or solid interfering with the construction of the cable.

### 4.3 Installation

Installing subsea cables is initiated by removing any wreckage out the direct way of the desired cable route, for example, with a pre-lay grapnel procedure. The installation of a subsea cable involves a dedicated cable laying vessel. The vessel uses a carousel cable container to load, transport and install the cable. The technical specification of each cable will define individual cable-handling limits, for instance, the bend radius limit will differ depending on the cable design and manufacturing

parameters. These must to be known to the installers when the installation methods are being predetermined. As offshore wind farms operate in harsh conditions, assembling these assets can be challenging. The installation process requires expertise and precision, and human error can ensue.





### 4.4 External Damage

Subsea cables can be installed across the seabed in a variety of ways. The cable can be buried under the seabed up to 3m below
the surface, however 2m is the expected depths for most assets. Where the cable cannot be protected through burial, it can be
shielded from external damage with concrete mattresses or weighed down with rock placement. The threats to a subsea cable
from external factors can include: the mobility of the seabed sediment, scouring, instability of the cable's placement due to
environmental conditions, and interaction with fishing gears and anchors. External damage can occur from instability from
surrounding conditions. This can be due to:

- extreme metocean (for example, wind and wave) conditions
- unexpected soil conditions
- interaction with crossing of pipes and cables
- insufficient scour protection

### 5. Conclusion

As the collated dataset continues to expand in quantity and reliable resources, the root causes and failure modes will provide
trends that can bridge gaps in the industry. This will reduce the volume of cable failures by encouraging and supporting
innovation to tackle key challenges. The correct interpretation of results can provide insight on lessons learned from past events
in cable failure without destroying the anonymity of those who participate.

The industry's push to favour data sharing and the continued growth of this data will help to inform of both the severity of the
challenges associated with cable failures and how this impacts costs, generation losses and other financial liabilities of current
and future technology development. The motive is to use this data to determine trends for operational offshore wind projects
and to demonstrate the level of failures that can occur in subsea cable systems. The potential that these trends have in raising
awareness and initiating discussion is of value to the industry and encourages scrutinised study of preceding events.

It is essential to continue to grow and develop data of such a sweeping industry matter like cable failure. This will allow better
analysis and will provide a better representation of the industry.

### 6. Author Contribution

The author was involved in the design of the database, collection, derivation and analysis of the data as part of ORE Catapult.



## 7. Competing Interests

245 The authors declare that they have no conflict of interest.

## 8. Disclaimer

Whilst the information contained in this report has been prepared and collated in good faith. Charlotte Strang-Moran makes no representation or warranty (express or implied) as to the accuracy or completeness of the information contained herein nor shall we be liable for any loss or damage resultant from reliance on same.

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
