# Peer review of "Subsea cable management: Failure trending for offshore wind"

_Wind Energy Science, 2020_

## Short Comment (SC1) · 31 Mar 2020

Line 49 - Creyke Beck is no longer being used, Dogger Bank A, B and C - https://doggerbank.com/downloads/Dogger-Bank-name-change.pdf

Line 171 - LCoE has been used as opposed to CFD values? Is there reasoning for this? Also for IACs has the position of the fault on the string been considered?

Section 4.3 - Does installation consider the transport and storage of the cables? Cables which have been onloaded/offloaded multiple times would appear to be at higher risk than where this has been limited, particularly with aluminium cored cables.

Line 220 - Is there a source for the 2-3m burial depth? This seems to be the top end for burial for IAC.

---

## Author Comment (AC1) · 11 May 2020

Hi Daniel, thank you for your comment on Wind Energy Science discussion forum.

Please see below, my responses regarding:

Line 49- Thank you, I will update this

Line 171- The wind farms included in the trend analysis cover a range of installation years, and therefore a mix of support mechanisms (power price + ROC's; and CfD's). For the purpose of this report the LCoE was the most fitting in terms of simplistic estimations of lost generation. This being said, using installation year LCOE based on industry cost reduction profile can understate the value of lost generation for some operating wind farms, and therefore the results can be used as a suggested base line

rather than an accurate hindcast.

Section 4.3 - Installation does account for transport and storage of cables, thank you for your input on this topic. Unfortunately, due to our collated data at this current point in time, we saw it better to categorise in one failure mode of installation. As more data is collated, it would be an excellent next step to include storage and transport as separate modes.

Line 220 - 2-3m burial depth is indeed the top end of depth. From engagement with stakeholders from the installation phase, it was concluded that cable can be buried under the seabed up to 3m below the surface (as the top end depth), however 2m is the expected top end depth for many assets. This perhaps isn't demonstrated correctly with my choice of wording, and I am happy to update the paper to ensure clarity.

---

## Referee Comment (RC1) · Anonymous Referee #1 · 19 May 2020

General Comments This paper discusses about the failure of subsea cables for offshore wind. It lacks substantial scientific literature review to place the reader into context. This is also particularly helpful for establishing the scientific contributions of a manuscript, which I do not see clear. Some figures are not properly described, the methodology is extremely short and it does not explain what was done exactly, and the results do not provide important outcomes.

Specific Comments - Literature review/Introduction. This needs to be noticeably enlarged to bring value to the paper. International organizations such as CIGRE and IEC have reports regarding cables failures (CIGREWorking Group B1.10. Update of Service Experience of HV Underground and Submarine Cables; CIGRE: Paris, France, 2009, CIGREWorking Group B1.21. Third-Party Damage to Underground and

Submarine Cables; CIGRE: Paris, France, 2009.), and some scientific publications are available as well in this particular topic (for instance, https://www.mdpi.com/1996-1073/12/14/2682). - Methodology. It needs to be described how the database was populated, indicate sources and parameters under study. If possible, it needs to be benchmarked with other databases. - Results. Statistical analysis is required. In Figure 2, there is no explanation about what needs to be understood for the different types of root causes. The database should lead to the estimation of the Mean Time Between Failures parameters, but this is never discussed in the article. In Figure 3, all required parameters must be indicated: assumed price of energy, type of cable failure, projects under analysis, etc. The failures need to be segregated between MV and HV, to show the influence of the voltage level over failures.

---

## Author Comment (AC2) · 20 May 2020

Thank you Anonymous Referee #1 for your feedback on Wind Energy Science discussion forum.

There is a lot of interest around failure of offshore wind subsea cables, both export and inter-array, and there are many more pieces of literature that discusses failure modes across HV and submarine cables. What is currently an interest area and this paper is highlighting, is the lack of data available from offshore wind farms. To truly understand the progression and development in offshore wind cable technologies, as well as understanding safer offshore fault-finding procedures the recording of causes of cable failure and lessons learned should be attained. Currently there is limited existing

data on cable failures and there has been no effective concept to benchmark them. The database currently contains data from both public sources, available on open sources online, and sources from our own engagement within the industry. It is important to highlight that this data is from a wide range of sources which is not the most effective way to collect information. However, as the cable sector has such limited data on failures – this is the only way it can be done. Stakeholder engagement clarified that there were reservations over whether a live, regularly-updated platform would provide ongoing value and longevity, but the industry is changing and data trend analysis is recognised as an effective method to bridge certain technical challenges in the industry. More opportunities are materialising for accurate data-sharing. We have been working towards better and more accurate ways to obtain our data so that we can investigate into more detail certain aspects in the cable sector.

Cigre's data is very useful and helpful in investigating cable failure across HV cable underground, inter-connectors and some offshore wind cabling. However, it does not provide data entirely investigating inter-array or export cables for offshore wind. As our data currently sits, we have provided export and inter-array cable finding in accumulated results due to the limited data. As we continue to advance our data sets we will be looking to improve they way in which the data can be presented.

The objective of the report is to open up discussions for data sharing and how the information can be displayed and trended while ensuring those involved that anonymity is kept. We are now looking at the development of the database to improve data sources and therefore the accuracy, expediency and effectiveness of the results. This will allow further investigation into such things as estimation of the mean time between failures and failure rates for each type of cable MV and HV, from the collection grid and transmission. As we begin to grow and develop our failure recording abilities we are indeed looking to do this.

---

## Referee Comment (RC2) · Anonymous Referee #2 · 18 Sep 2020

The topic of the paper is of importance for the wind energy community. Submarine cables are a very important component and it can have significant impact on the generation. The initiative of collecting such data is laudable and of utility to the community. In this sense, is very important to communicate and disseminate the results. From a scientific perspective, the paper does not manage to highlight its contributions. The methodology section is very short and, in general, the paper is rather descriptive than scientific. While I find it very interesting and important topic, I think it will be more appropriate to be published as a report rather than a scientific paper.

---

## Author Comment (AC3) · 24 Sep 2020

Thank you, Anonymous Referee #2 for your feedback on Wind Energy Science discussion forum. I understand that the paper does not provide contributions as a scientific paper. After positive feedback from the conference the next step was to input a paper. Due to the level of data able to be published at this time (for reasons of confidentiality), it is not possible to elaborate on the methodologies further and provide more technical input.

Thank you for acknowledging the work as an important topic, and I agree, it will be more appropriate to be published as a report rather than a scientific paper. I will look to publishing as a technical paper in the future if there is sufficient technical data which

is able to be publicly released.

---

## Author Comment (AC4) · 24 Sep 2020

I acknowledge that I cannot make all the changes required from the referees. The following are highlighted areas from comments that I can amend: • Line 49 - Creyke Beck is no longer being used, Dogger Bank A, B and C • Line 171 – add explanation to why LCoE has been chosen instead of CfD • Line 220 – strengthen explanation of why 2-3m burial depth is mentioned • Add explanation to why current available open source data is not representative of the industry

I understand that the paper does not provide contributions as a scientific paper. After positive feedback from the conference the next step was to input a paper. Due to the level of data able to be published at this time, it is not possible to elaborate the

methodologies further and provide more technical input. Therefore, it may be more suited for a different journal, or published independently.

Additional changes: • Correcting/updating terminology for root causes and failure modes.